# A Lean Dataset for International Math Olympiad: Small Steps towards Writing Math Proofs for Hard Problems

**Roozbeh Yousefzadeh**                                                      *yousefzadeh.roozbeh@huawei.com*
*Huawei Hong Kong Research Center*

**Xuenan Cao**                                                              *xuenancao@cuhk.edu.hk*
*Department of Cultural and Religous Studies, The Chinese University of Hong Kong*

**Azim Ospanov**                                                            *aospanov9@cse.cuhk.edu.hk*
*Huawei Hong Kong Research Center*
*Department of Computer Science & Engineering, The Chinese University of Hong Kong*

**Reviewed on OpenReview:** *https://openreview.net/forum?id=CrKMqRAhBo*

## Abstract

Using AI to write formal proofs for mathematical problems is a challenging task that has seen some advancements in recent years. Automated systems such as Lean can verify the correctness of proofs written in formal language, yet writing the proofs in formal language can be challenging for humans and machines. The miniF2F benchmark has 20 IMO problems in its test set, yet formal proofs are available only for 6 of these problems (3 of which are only written by mathematicians). The model with best accuracy can only prove 2 of these 20 IMO problems, from 1950s and 60s, while its training set is a secret. In this work, we write complete, original formal proofs for the remaining IMO problems in Lean along with 3 extra problems from IMO 2022 and 2023. This effort expands the availability of proof currently in the public domain by creating 5,880 lines of Lean proof. The goal of the paper is to pave the way for developing AI models that can automatically write the formal proofs for all the IMO problems in miniF2F and beyond by providing an evaluation benchmark. In this pursuit, we devise a method to decompose the proofs of these problems into their building blocks, constructing a dataset of 1,329 lemmas with more than 40k lines of Lean code. These lemmas are not trivial, yet they are approachable, providing the opportunity to evaluate and diagnose the failures and successes of AI models. We evaluate the ability of the SOTA LLMs on our dataset and analyze their success and failure modes from different perspectives. Our dataset and code is available at: `https://github.com/roozbeh-yz/IMO-Steps`.

## 1 Introduction

Machine learning methods have made significant progress in many different domains. This progress can often be described as the automation of learning from data. The availability of high quality data is crucial for the community to develop better models and methods. ImageNet (Deng et al., 2009), for example, played a significant role in the development of ML methods in computer vision. More recent models such as OpenAI's o1 and Llama rely on training sets purportedly almost as large as the entire contents of the Web. On the other hand, the failures of these models on topics such as math are often explained via the lack of high-quality materials on those topics in their training set. This issue is partially addressed by curation of relevant data, e.g., Lightman et al. (2023) recruits human labelers and AlphaGeometry (Trinh et al., 2024) generates synthetic data. For the topic of automated theorem proving in formal language, high-quality data is so scarce, to the extent that the formal proofs are not available for almost half of the problems in the miniF2F dataset, the benchmark of the community. So, when automated methods fail to write a correct proof for the problems in the benchmark dataset, it is not clear where they fail, and what the correct answer looks like. In this work,

we contribute the complete, original, formal proofs in Lean language for all the 14 IMO problems in the test set of miniF2F that do not have a formal proof publicly available. Moreover, we study 13 IMO problems in detail, breaking them down into their building blocks, leading to a dataset of 1,330 lemmas. We then evaluate the ability of SOTA LLMs on our dataset. The results reveal the abilities and inabilities of these models in proving the building blocks of IMO problems.

Solving mathematical problems is a challenging task for machine learning. There are many motivations for doing math with AI as we will discuss further in the Appendix J. One motivation is to continue creating machines that can outperform humans in a given task, such as in the game of chess (Pandolfini, 1997) and two decades later in the game of Go (Silver et al., 2017). Currently, there is a 10-million dollar prize for an AI system that can participate in the International Math Olympiad and win a gold medal: the AIMO Prize. There is also the IMO Grand Challenge which specifically requires the model to write its proof in Lean language. Lean is a functional programming language and a proof assistant (de Moura et al., 2015) used mostly by mathematicians to write and verify mathematical proofs. The Lean system makes it possible to verify the correctness of a proof almost instantly. This has given rise to a field known as formal mathematics. Lean is also used by Amazon AWS for formal verification of software (Cutler et al., 2024).

Our approach in this paper is to provide a stepping stone for proving IMO problems in the miniF2F dataset and beyond. With the help of our dataset and the complete proofs that we contribute to the test set of miniF2F benchmark, we hope to enable researchers to identify and better understand the weaknesses of their models. Consider the case of a math teacher who encounters a student who cannot solve a hard problem. The teacher might ask the student to solve easier problems first, such as the building blocks of that hard problem. How much the student can solve the easier problems will help the teacher evaluate the student's strength and weaknesses. Likewise, the building blocks we provide in our dataset can be considered stepping stones for evaluation purposes. For further illustration, we can also break down the evaluation into scenarios such as the following: 1- The student cannot provide a correct answer for any of the easy problems. The teacher might start asking even simpler questions. 2- The student can provide a correct answer for some of the easy problems. The teacher may be able to identify the underlying weaknesses of the student. Perhaps the student has not been exposed to certain subjects. Or, maybe the student has not learned certain reasoning methods. 3- The student can solve all of the easy problems. Then the teacher might try to teach the student how to put the answers for the easy problems together to compose the answer for the more complex problems.

The lemmas that we provide in our dataset are analogous to the easy problems that the teacher uses in the example above. Therefore, our lemmas are designed to better evaluate the models and provide a way to diagnose their failures. We do not suggest that people should fine-tune their models on the complete proofs that we provide. Rather, we suggest creating a clear separation between the training set of the models and the test set of miniF2F. By making these proofs public, we anticipate the community will move towards better evaluation practices.

## 1.1 IMO: an unknown test set that gets harder every year

Challenging an AI model to participate in an IMO and win a gold medal has certain implications, especially regarding the evaluation. First, the problems presented in an IMO are revealed to participants on the day of the exam. Doing well in the IMO requires a mastery of the subjects presented in it. Merely studying the problems from the previous iterations of the Olympiad is not sufficient as the problems tend to become harder year by year. Second, the problems that appear in IMO go through a selection process that ensures some degree of novelty. As a result, these problems are usually not present in text books. It follows that direct retrieval from the textbook solutions is usually not sufficient to prove the new IMO problems. Third, the exam is timed. A participant must complete the exam within 9 hours.

These conditions, in the language of machine learning, implies being evaluated on an unknown test set with samples that are more challenging than the samples in the training set. In the machine learning literature, however, having standard benchmarks with fixed and accessible test sets is essential for a large community to be able to work towards common goals. This approach of using a fixed test set has worked reasonably well in some ML fields (Recht et al., 2019), mostly because the training sets of the models are fixed, too, and in creating these datasets, specific procedures are followed to ensure samples from training sets do not appear in

the test sets. However, with the rise of large language models, it has become increasingly difficult to evaluate how much of a model's success is merely direct retrieval of the correct answer from its training set. Although for solving mathematical problems, the miniF2F (Zheng et al., 2021) dataset is the common benchmark with a fixed test set, the training set of LLMs is sometimes unknown; when it is known, it includes all the relevant content available on the Internet, (e.g., all the available code on Github, arXiv, Stack Exchange, and elsewhere), which is an enormous corpus, hard to search through and compare against the test sets. When evaluating LLMs developed on such training sets, we often use a fixed test set. But, IMO evaluates on an unknown test set that gets harder every year.

To prepare an AI model that can do well in IMO, we need to rely on rigorous evaluation techniques that can tell us whether the model can generalize well on an unknown test set. In some applications, the distinction between generalization and retrieval might be besides the point. As long as a model performs the task correctly, one might not be too keen to evaluate what percentage of its correct answers is a direct retrieval from its training set. However, when we are developing a model that is intended to be evaluated on an unknown test set, we will need to rely on clearer evaluation methods.

## 1.2 Where we are

Let us now take a closer look at the current benchmark: miniF2F dataset. This dataset consists of mathematical theorems with varying degrees of difficulty. Out of the 244 theorems in its test set, 20 of them are IMO problems, and the difficulty levels of those IMO problems also vary since they are taken from years 1959 to 2019. In terms of complexity, some of the theorems in the dataset can be proved with one line of formal proof while others may need more than a thousand lines of formal proof (see Table 1). Success in solving a high percentage of the less complex problems is not the same as success in solving the same percentage of the more complex problems. Hence, merely reporting the accuracy by the percentage of the test set is not insightful enough.

LEGO-Prover (Wang et al., 2024), the current peer-reviewed state of the art method on miniF2F, is able to prove only one of the IMO problems in this dataset: the one from 1959, which is the first problem of the first IMO competition implying that it is the easiest of all IMO problems proved in less than 10 lines of code. Other ML methods have proved more of the IMO problems in the miniF2F, e.g., the HTPS method by Lample et al. (2022) claims proving 10 IMO problems (not all from the miniF2F), but those problems are either from 1960s or the ones for more recent years have a simplified or erroneous problem statement. The method by Polu et al. (2022) also proves two adapted (simplified) IMO problems.

## 1.3 Our approach

For the majority of the IMO problems in miniF2F, even a human-written formal proof is not publicly available. In this work, we look into these specific problems along with 3 more challenging problems from IMO 2022 and 2023.

We provide the formal proofs for these problems and further evaluate the ability of o3-mini (OpenAI, 2025) in writing these proofs. We decompose each of the formal proof steps into small lemmas and report that o3-mini cannot write a correct formal proof for almost all of the lemmas. Writing the formal proof for these problems can be viewed from the perspective of planning and execution: one would need to prove a certain chain of lemmas (execution), and put those lemmas together (planning) in a coherent way to compose the final proof. The planning aspect of writing formal proofs is challenging, and in the literature, often the plan, i.e., the proof sketch or the *approach*, is taken from the informal proofs written by humans (Jiang et al., 2022).

We diagnose the failures of o3-mini by evaluating and labeling its proofs manually. We further guide the model by pointing out its mistakes. Giving feedback to the model is mostly unfruitful, especially on the topics that o3-mini seems to be less educated.

## 1.4 Our contributions

1. We provide the formal proofs for 17 IMO problems that do not have a formal proof in Lean. 14 of these problems are from the miniF2F dataset. The other three are from IMO 2022 and 2023. We

specifically chose the problems that do not have a formal Lean proof in the public domain. Our proofs are written in Lean 4 and consist of 5,790 lines of code.

2. We decompose the proofs of 13 IMO problems into small lemmas, creating a dataset of 1,329 lemmas that are still challenging for AI models, yet they are easier and more approachable.

3. We evaluate the ability of the stat-of-the-art LLMs (Goedel-Prover (Lin et al., 2025), DeepSeek-Prover-V1.5-RL (Xin et al., 2024b), and OpenAi's o3-mini) in writing the formal proofs for the lemmas in on our dataset. We further perform a human review of o3-mini's informal and formal proofs after 10 rounds of expert feedback. We categorize the issues in the incorrect proofs providing deep insights about what goes wrong. We also analyze the correct proofs observing that they are limited by length.

## 2 Dataset of approachable lemmas: Stepping stone towards solving challenging IMO problems

Table 1 describes the 13 IMO problems that we study in this paper. The concepts used in these problems are described in further detail in Table B1. In appendices I and H, we provide the datasheet and license information about our dataset.

Note that we provide the Lean proofs for all the IMO problems in the test set of miniF2F, but we limit our study to a subset of these problems listed in Table 1. Table C2 provides information about the proof of all the problems in our dataset including the ones that we do not perform lemma decomposition. Additionally, Table D3 provides information about the lengths of the proofs for the lemmas in our proposed dataset.

Table 1: IMO problems formalized and studied in this paper

| # | Year | Problem | Topic | in miniF2F | Lean proof publicly available | # of lemmas | # of lines of Lean4 proof |
|---|---|---|---|---|---|---|---|
| 1 | 1959 | p1 | number theory | Yes | Yes | 4 | 9 |
| 2 | 1960 | p2 | algebra | Yes | Yes | 9 | 40 |
| 3 | 1962 | p2 | algebra | Yes | No | 14 | 60 |
| 4 | 1964 | p2 | algebra | Yes | Yes | 9 | 50 |
| 5 | 1965 | p2 | algebra | Yes | No | 73 | 210 |
| 6 | 1983 | p6 | algebra | Yes | No | 53 | 180 |
| 7 | 1984 | p6 | number theory | Yes | No | 64 | 380 |
| 8 | 1985 | p6 | number theory | Yes | No | 427 | 1,310 |
| 9 | 1992 | p1 | number theory | Yes | No | 91 | 480 |
| 10 | 1997 | p5 | number theory | Yes | No | 122 | 390 |
| 11 | 2022 | p2 | algebra | No | No | 61 | 260 |
| 12 | 2022 | p5 | number theory | No | No | 265 | 640 |
| 13 | 2023 | p4 | number theory | No | No | 137 | 450 |
| total | | | | | | 1,329 | 4,459 |

Making these formal proofs accessible to the research community can be helpful in different ways. First, every one, including non-mathematicians, will know the work it takes to write a formal proof for these problems. Second, these formal proofs can be used to identify possible mistakes or shortcomings in the informal proofs that might be circulating. For example, Bubeck et al. (2023) reports that GPT-4 can write a correct informal proof for IMO 2022 P2. However, when we write the formal proof for the same problem, it becomes obvious that the informal proof was missing a crucial step. Third, some of the formal proofs in the literature simplify the original IMO problem by altering the problem statement, but we remain faithful to the original problem statements of IMO. For example, Xin et al. (2024a) showcases the proof for two shortlisted IMO problems. In the first problem, IMO 2009 P3, they have a mistake in the problem statement, and the proof merely uses

the error to prove False. For the second problem, IMO 2016 P5, the statement is altered from proving "there exists infinitely many solutions" to "there exists a solution".

## 2.1 Decomposition of the proofs into lemmas

In the next step, we decompose the proofs for each of these problems into smaller lemmas. For example, the proof might involve showing that given a set of constraints, natural number $p$ cannot be larger than 3. Then, use another series of deductions to show that $p$ cannot be smaller than 2. And then use these two results to conclude that $p$ must be either 2 or 3. Section 4 provides a more detailed discussion about decomposition of lemmas and an upper bound on how many proofs can be extracted from $n$ lines of proof. Moreover, Table D3 provides some statistics about the length of the proofs for the lemmas in our dataset.

All of the 1,329 lemmas in our dataset have a formal proof in Lean 4, version 4.17. Some of the easier lemmas are as the following:

- $q$ and $r$ are integers and their product is 11, prove that $q$ is either $\pm 1$ or $\pm 11$

- $p$ is a natural number greater than 4, prove that rational number $\frac{p}{(p-1)} \leq 4/3$

- $k$ is natural number greater than 4, prove that $k < 2^{(k-2)}$

Our library of lemmas also includes harder problems such as proving injectivity of functions, points about rational numbers, casting, divisibility. Here are some examples:

- $x$ and $y$ are positive natural numbers where $y < x$, and we have $(x^y)^2 = y^x$, prove that $y^2$ divides $x$

- $a$ and $b$ are natural numbers and $p$ is a prime number where $a^p = b! + p$ and $p \leq b$, prove that $p$ divides $a$

- $b$ is a natural number and $p$ is a prime number, where $p \leq b$ and $p^p = b! + p$, prove that $p$ must be less than 5

- $f$ is a real-valued function that is strictly monotonic and upper bounded by $a$, prove that $a$ cannot be in the range of $f$.

The lemmas that are harder to solve are also broken into smaller pieces, so the dataset includes both the hard version of such lemmas plus the smaller lemmas needed to prove them.

## 3 Evaluating the ability of LLMs in proving our lemmas

We first evaluate o3-mini in great depth, then we analyze the latest SOTA models on our dataset. Finally, we analyze the success and failure cases of the LLMs.

## 3.1 o3-mini's ability in formal and natural language

We prompt o3-mini with a statement requesting it to write a formal proof for the problem, instructing it to think step by step, explain the theorem, the approach, and the proof, following prompting techniques suggested by Kojima et al. (2022). We also provide the formal Lean statement of the problems in the prompt so that issues about the formalization of the theorem statement will not arise. Furthermore, we verify that o3-mini's explanation of the problems in natural language matches the provided Lean statements. Table 2 shows the evaluation of the responses of o3-mini in natural language (NL) and in Lean. The last column of this table evaluates whether there is a match between the proof written in NL and the one in Lean. A LLM may describe a proof in NL, but follow a different approach when writing the proof in Lean.

Table 2: Analyzing the o3-mini's responses to our lemmas

| # | Problem | # of lemmas | Correct proof in NL | Correct proof in Lean | Match between NL and Lean |
|---|---------|-------------|---------------------|----------------------|---------------------------|
| 1 | 1959-p1 | 4 | 100% | 50.0% | 100% |
| 2 | 1960-p2 | 9 | 55.6% | 11.1% | 100% |
| 3 | 1962-p2 | 14 | 92.9% | 42.9% | 100% |
| 4 | 1964-p2 | 9 | 77.8% | 33.3% | 100% |
| 5 | 1965-p2 | 73 | 97.3% | 16.4% | 100% |
| 6 | 1983-p6 | 53 | 64.2% | 34.0% | 100% |
| 7 | 1984-p6 | 64 | 73.4% | 20.3% | 100% |
| 8 | 1985-p6 | 427 | 75.2% | 19.7% | 95.6% |
| 9 | 1992-p1 | 91 | 87.6% | 27.5% | 100% |
| 10 | 1997-p5 | 122 | 69.7% | 24.6% | 100% |
| 11 | 2022-p2 | 61 | 77.0% | 41.0% | 100% |
| 12 | 2022-p5 | 265 | 63.4% | 22.6% | 92.8% |
| 13 | 2023-p4 | 137 | 88.3% | 27.0% | 92.7% |
| total | | 1,329 | 75.5% | 23.8% | 96.4% |

We provide up to 10 rounds of feedback to o3-mini in the instances that it does not write a correct proof. The feedback is automatically generated using REPL[1], a library in Lean which summarizes the errors when compiling a Lean file. The results are presented in Table 3 demonstrating the effectiveness of our automated feedback system in improving the accuracy of o3-mini.

Table 3: The effect of automatic feedback loop on the accuracy of o3-mini

| # | problem | % correct zero shot | % correct after 5 rounds of feedback | % correct after 10 rounds of feedback |
|---|---------|---------------------|--------------------------------------|---------------------------------------|
| 1 | 1959-p1 | 50.0% | 50.0% | 50.0% |
| 2 | 1960-p2 | 0.0% | 11.1% | 11.1% |
| 3 | 1962-p2 | 7.1% | 21.4% | 42.9% |
| 4 | 1964-p2 | 22.2% | 33.3% | 33.3% |
| 5 | 1965-p2 | 0.0% | 8.2% | 16.4% |
| 6 | 1983-p6 | 9.4% | 20.8% | 34.0% |
| 7 | 1984-p6 | 3.1% | 14.1% | 20.3% |
| 8 | 1985-p6 | 4.9% | 16.2% | 19.7% |
| 9 | 1992-p1 | 5.5% | 16.5% | 27.5% |
| 10 | 1997-p5 | 4.9% | 16.4% | 24.6% |
| 11 | 2022-p2 | 16.4% | 36.1% | 41.0% |
| 12 | 2022-p5 | 4.5% | 18.1% | 22.6% |
| 13 | 2023-p4 | 10.2% | 24.8% | 27.0% |
| total | | 6.0% | 18.4% | 23.8% |

Moreover, Table 4 shows a detailed evaluation of the formal proofs written by o3-mini at its final attempt. If a proof is completely correct and can pass the Lean system, it will be counted in the column "No error". Otherwise, a proof may have one or more of the following issues: hallucinations, wrong approach, wrong implementation, incomplete proof, and/or minor errors.

"Hallucination" refers to cases where the proof contains one or more lemmas or tactics that do not exist in the Mathlib library and it is also not defined separately in the proof. This happens more often when the

---

[1]https://github.com/leanprover-community/repl

model is uneducated on a topic. In such cases, LLM may assume that there is one or a few magic lemmas in the Mathlib library that can prove the given lemma outright or in a few steps.

"Wrong approach" refers to cases where the approach to prove a lemma is wrong. This means that either some of the individual proof steps are wrong or the steps do not logically lead to what the model wants to prove.

"Wrong implementation" refers to cases where the model does not use the correct procedures or it does not call the correct lemmas from the Mathlib library.

"Incomplete proof" refers to cases where the proof is explicitly or implicitly incomplete. An explicitly incomplete proof contains one or more *sorry* statements, whereas an implicitly incomplete proof does not contain such statements but is unfinished.

Finally, "minor error" refers to cases where the model uses a correct lemma or tactic from the Mathlib library but it does not provide the correct arguments to it, or it misses some of the arguments.

A given proof may have several lines of code, and many of these issues may arise. For example, a proof may have a few hallucinations, a wrong implementation, and also a few minor mistakes.

Table 4: Analyzing the errors in the Lean proofs written by o3-mini

| # | Problem | No error | One or more types of error | | | | |
|---|---------|----------|---------------|------------------|----------------------|-------------------|----------------|
| | | | Hallucination | Wrong approach | Wrong implementation | Incomplete proof | Minor error |
| 1 | 1959-p1 | 50.0% | 50.0% | 0.0% | 50.0% | 0.0% | 50.0% |
| 2 | 1960-p2 | 11.1% | 33.3% | 44.4% | 11.1% | 0.0% | 88.9% |
| 3 | 1962-p2 | 42.9% | 7.1% | 7.1% | 28.6% | 0.0% | 57.1% |
| 4 | 1964-p2 | 33.3% | 22.2% | 22.2% | 0.0% | 11.1% | 44.4% |
| 5 | 1965-p2 | 16.4% | 6.8% | 2.7% | 26.0% | 1.4% | 83.6% |
| 6 | 1983-p6 | 34.0% | 0.0% | 35.8% | 35.8% | 3.8% | 60.4% |
| 7 | 1984-p6 | 20.3% | 0.0% | 26.6% | 20.3% | 3.1% | 79.7% |
| 8 | 1985-p6 | 19.7% | 0.5% | 18.0% | 53.4% | 14.8% | 79.9% |
| 9 | 1992-p1 | 27.5% | 1.1% | 12.1% | 48.4% | 1.1% | 72.5% |
| 10 | 1997-p5 | 24.6% | 4.1% | 30.3% | 31.1% | 6.6% | 75.4% |
| 11 | 2022-p2 | 41.0% | 6.6% | 23.0% | 23.0% | 0.0% | 59.0% |
| 12 | 2022-p5 | 22.6% | 2.3% | 32.1% | 58.1% | 9.4% | 74.3% |
| 13 | 2023-p4 | 27.0% | 0.0% | 1.5% | 63.5% | 12.4% | 73.0% |
| total | | 23.8% | 2.3% | 20.4% | 46.9% | 9.0% | 75.1% |

### 3.2 Performance of SOTA models on the dataset

We then evaluate the state-of-the-arts models in the literature. We consider Goedel-Prover as it has the highest accuracy on the miniF2F test set. We also consider DeepSeek Prover 1.5 which is the base of the Goedel-Prover and it used to be the model with the highest accuracy before the release of Goedel-Prover. Finally, we consider ReProver, with and without retrieval, which is an established model in the literature with an open-source training set. We note that both Goedel-Prover and DeepSeek Prover do not have an open-source training set while ReProver has. Table 5 shows the accuracy of these models on our lemmas based on the IMO problem where the lemmas are extracted from.

The model with the best accuracy on our lemmas turns out to be the DeepSeek-Prover with accuracy of 39.3%. Although Goedel-Prover has a higher accuracy on miniF2F test set, it shows a lower performance on our dataset, proving 18 lemmas fewer than DeepSeek. At the same time, ReProver with retrieval proves 83 lemmas fewer than DeepSeek corresponding to accuracy of 33%.

o3-mini proves a total of 316 lemmas corresponding to accuracy of 23.8%. This accuracy is lower than the accuracy of the other three LLMs, yet it seems impressive given that, unlike the previous models, o3-mini is

Table 5: Comparing performance of SOTA models on our lemmas dataset (blue shows the IMO problem that a model has the best accuracy on its extracted lemmas, and red marks the IMO problem that a model has the worst accuracy on its extracted lemmas. underline marks the model with the highest accuracy on the lemmas extracted from each IMO problem.)

| Problem | # of lemmas | DeepSeek Prover-v1.5-RL (@32) | Goedel-Prover (@32) | ReProver retrieval ✗ | ReProver retrieval ✓ | o3-mini (with 10 e.f.) |
|---------|-------------|-------------------------------|---------------------|----------------------|---------------------|------------------------|
| 1959-p1 | 4 | 3 (75.0%) | 2 (50.0%) | 2 (50.0%) | 2 (50.0%) | 2 (50.0%) |
| 1960-p2 | 9 | 7 (77.8%) | 6 (66.7%) | 3 (33.3%) | 4 (44.4%) | 1 (11.1%) |
| 1962-p2 | 14 | 13 (92.9%) | 12 (85.7%) | 7 (50.0%) | 8 (57.1%) | 6 (42.9%) |
| 1964-p2 | 9 | 5 (55.6%) | 5 (55.6%) | 5 (55.6%) | 5 (55.6%) | 3 (33.3%) |
| 1965-p2 | 73 | 48 (65.8%) | 47 (64.4%) | 47 (64.4%) | 46 (63.0%) | 12 (16.4%) |
| 1983-p6 | 53 | 25 (47.2%) | 32 (60.4%) | 28 (52.8%) | 29 (54.7%) | 18 (34.0%) |
| 1984-p6 | 64 | 31 (50.0%) | 33 (51.6%) | 25 (39.1%) | 24 (37.5%) | 13 (20.3%) |
| 1985-p6 | 427 | 116 (27.2%) | 116 (27.2%) | 89 (20.8%) | 89 (20.8%) | 84 (19.7%) |
| 1992-p1 | 91 | 48 (52.7%) | 54 (59.3%) | 35 (38.5%) | 34 (37.4%) | 25 (27.5%) |
| 1997-p5 | 122 | 51 (41.8%) | 49 (40.2%) | 48 (39.3%) | 51 (41.8%) | 30 (24.6%) |
| 2022-p2 | 61 | 34 (55.7%) | 30 (49.2%) | 24 (39.3%) | 25 (41.0%) | 25 (41.0%) |
| 2022-p5 | 265 | 89 (33.6%) | 76 (28.7%) | 80 (30.2%) | 77 (29.1%) | 60 (22.6%) |
| 2023-p4 | 137 | 52 (38.0%) | 41 (29.9%) | 43 (31.4%) | 45 (32.8%) | 37 (27.0%) |
| Total | 1,329 | 522 (39.3%) | 504 (37.9%) | 436 (32.8%) | 439 (33.0%) | 316 (23.8 %) |

a general purpose model, not specifically designed for proving mathematical theorems in Lean language. This indicates a high potential for o3-mini to achieve SOTA accuracy on our dataset and on miniF2F after being fine-tuned on appropriate data such as the one used for fine-tuning Goedel Prover.

Additionally, we evaluated the accuracy of Qwen Math 2.5 (Yang et al., 2024) and Llama 3.1 (Dubey et al., 2024) on our lemmas in formal language. Qwen Math 2.5 was able to prove only 1 lemma while Llama failed to prove any of the lemmas. We note that these two models are not designed specifically for proving theorems in formal language, nevertheless, their ability for automated theorem proving significantly lag behind o3-mini. The complete results are shown in Table E4.

### 3.3 Analyzing the proof lengths

Here, we look at the patterns in the correct proofs written by the LLMs, specifically with regard to the proof lengths. Figure 1 shows the distribution of the length of correct proofs written by LLMs for the lemmas in our dataset, and Table 6 shows the accuracy of each LLM in proving the subsets of lemmas for various ranges of proof lengths. Out of the 1,329 lemmas in our dataset, Goedel Prover can prove 504 lemmas correctly. What is common among these 504 lemmas is that they all have short proofs, majority of them 1 liners using the automatic Lean solvers such as *nlinarith*. The longest correct proof that Goedel Prover has written for our lemmas has 15 lines and the longest proof written by DeepSeek-Prover has 12 lines. Surprisingly, the longest correct proof is written by o3-mini consisting of 34 lines.

For the lemmas that Goedel Prover and DeepSeek Prover can correctly prove, the average length of the proof is less than 3 lines, and the maximum length of 15 and 12 lines are several standard deviations away from that mean (See Appendix F for further details). Hence, writing a correct proof of 15 or 12 lines is more of an anomaly for Goedel and DeepSeek. Overall, one can conclude that the ability of these two LLMs in writing correct proofs seems to be limited only to short proofs. These observations suggest a low capability for planning and reasoning in these LLMs as writing a long proof requires devising an elaborate plan as a prerequisite.

On the other hand, o3-mini shows a higher capability in writing correct long proofs in Lean as shown in Figure 1. While o3-mini fails to write the correct proof for most of the 1-liners, it does considerably better than Goedel and DeepSeek in writing sensible long proofs in Lean. The average length of the correct proofs that o3-mini writes is 7.6 lines. Note that in Table 6, the proof lengths are based on the proofs in our dataset which tend to be shorter than the proofs written by LLMs. Based on those lengths, the accuracy of Goedel and DeepSeek is 0% for all the lemmas with proofs longer than 10 lines. o3-mini is the only model that proves some of such lemmas, achieving an accuracy of 5.8% for lemmas with proof length of 11 to 15 and an accuracy of 1.9% for lemmas with proof length of 26 to 100 lines. For lemmas with proof length of 6 to 11 lines, the best accuracy is 46% by Goedel, and for lemmas with proof length of 6 to 10 lines, the best accuracy is 10% by o3-mini. The only type of lemmas that DeepSeek performs better are the lemmas that have a proof of 1 or 2 lines. Even for those lemmas with very short proofs, none of the models can achieve a near perfect accuracy. All these observations hints at opportunities for improving these models.

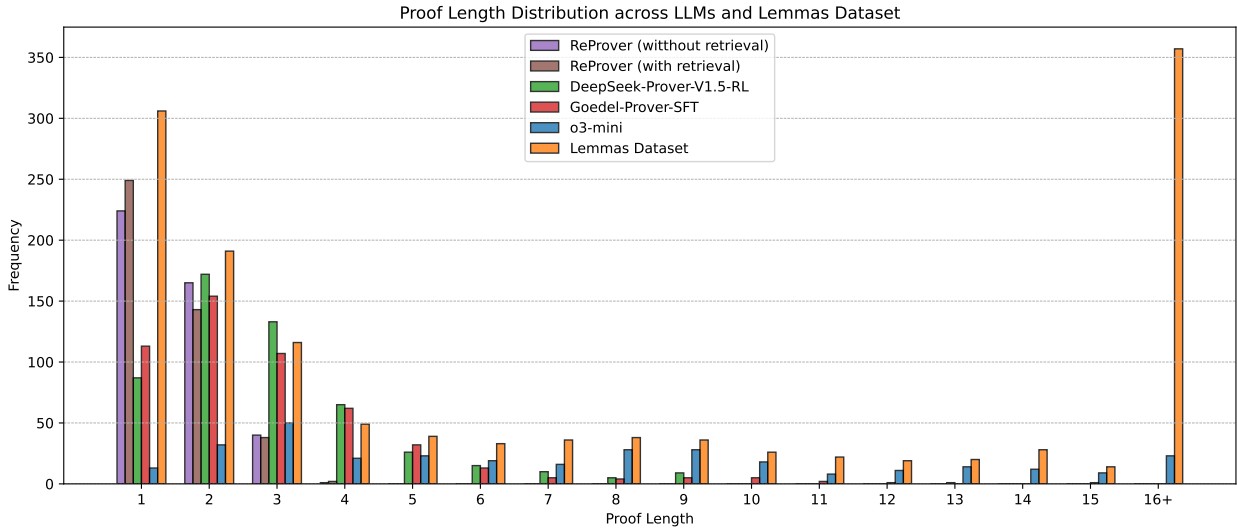

Figure 1: Distribution of the length of correct proofs written by LLMs. The histogram also includes the distribution of human-written proof lengths for comparison.

Table 6: Accuracy of models on our lemmas based on the length of proofs

| range of proof length | 1-2 | 3-5 | 6-10 | 11-15 | 16-25 | 26-100 | 101-298 |
|---|---|---|---|---|---|---|---|
| # of lemmas | 497 | 204 | 169 | 103 | 128 | 210 | 18 |
| Deepseek-Prover-v1.5-RL | 84.3% | 42.6% | 8.9% | 0.0% | 0.0% | 0.0% | 0.0% |
| Goedel-Prover | 79.5% | 46.1% | 8.9% | 0.0% | 0.0% | 0.0% | 0.0% |
| ReProver retrieval ✗ | 81.7% | 11.8% | 0.0% | 0.0% | 0.0% | 0.0% | 0.0% |
| ReProver retrieval ✓ | 82.9% | 9.8% | 0.0% | 0.0% | 0.0% | 0.0% | 0.0% |
| o3-mini | 49.7% | 24.0% | 10.1% | 5.8% | 0.0% | 1.9% | 0.0% |

We also observe that many of the correct proofs written by Goedel-Prover and DeepSeek-Prover contain a considerable number of irrelevant lines that can be purged from the proofs. Appendix G shows two examples of such proofs where large number of lines are used to prove irrelevant hypothesis or to repeat what is already proved or given. We refer to those proofs as bloated, and such irrelevant lines are not counted in the numbers reported in Figure 1 and Appendix F.

# 4 Decomposing a proof into its building blocks and extracting new lemmas

Here, we explain the procedures for lemma decomposition, and also derive upper bounds about the number of lemmas that can be extracted from an existing proof. The lemmas in our dataset were created by breaking down the proofs of the IMO problems. These proofs usually have specific structures that we exploit when breaking them down. First, we explain, how we exploit the structure of the proofs. Later, we will consider an extreme case and derive an upper bound on the number of lemmas that can be extracted from an $n$ line proof.

## 4.1 Breaking down the structure of a proof

Let us consider a theorem with $n$ lines of proof where $n > 3$. This proof may contain $k$ intermediate hypotheses that are proved one after another prior to proving the main statement of the theorem. These $k$ hypotheses, each have a proof of their own, and each may have intermediate hypotheses of their own, as well. Here, we only explain one level of breaking down the hierarchy.

Let us consider part of the proof for IMO 1997 P5 as an example. Here is the lemma:

$$x \, y : N, \quad h_0 : 0 < x \, \wedge \, 0 < y, \quad h_1 : (x^y)^2 = y^x, \quad h_2 : y < x, \quad := \quad y^2 | x,$$

First, we can consider each of the intermediate hypotheses and define them as a separate lemma. The intermediate hypotheses in the proof are:

- Lemma 1: prove $y^2 < x$

- Lemma 2: prove $2 * y^2 < x$

- Lemma 3: prove prime factorization of $(y^2)^x <=$ prime factorization of $x^x$

- Lemma 4: $(y^2)^x$ divides $x^x$

Each of these steps can be taken as a lemma.

Moreover, new lemmas can be constructed by the addition of the above lemmas to the hypothesis space of the original theorem.

- Lemma 5: grant lemma 1 to the original problem and prove $(y^2)^x$ divides $x^x$

- Lemma 6: grant lemmas 1,2 to the original problem and prove $(y^2)^x$ divides $x^x$

- Lemma 7: grant lemmas 1,2,3 to the original problem and prove $(y^2)^x$ divides $x^x$

- Lemma 8: grant lemmas 1,2,3,4 to the original problem and prove $(y^2)^x$ divides $x^x$

This latter process leads to new lemmas with proofs easier than the proof of the original problem. For example, the proof for lemma 5 would need to still prove lemmas 2,3,4 and then prove the ultimate goal. The proof for lemma 6 is shorter/easier than the proof for lemma 5 while it is longer than the proof for lemma 7, and so on.

Overall, these will lead to $2 * k$ lemmas. Each of these lemmas may also be broken down if they have proofs that are longer than a certain threshold.

In an extreme case, a proof may have $k = (n-1)/3$ intermediate hypothesis (each of them with 1 line of statement and 2 lines of proof leading to the 3 in the denominator) which would lead to $2/3 * (n-1)$ extracted lemmas. For example, in this setting, a proof with 4 intermediate hypothesis may have $4 * 3 + 1 = 13$ lines. And that leads to $2/3 * (13 - 1) = 8$ extracted lemmas.

### 4.2 Case of a proof with no intermediate structure

Now, we look at an extreme case of a proof with $n$ lines that does not have any specific structure, and we also assume that the $n$ lines of proof do not have any intermediate hypotheses. In such a case, we perform a forward path and two rounds of backward paths on the proof.

The *forward path* would start with the first line of the proof, apply the line, obtain the new proof state, and define that as a new lemma. The proof for this new lemma is the $(n-1)$ lines of the proof that remain. Continuing this way, the forward path can extract $(n-2)$ lemmas where each lemma has at least 2 lines of proof. Here, a minimum 2 lines of proof can be used as a threshold on the easiness of lemmas.

The *first backward path* would take each pair of consecutive lines in the proof, take the changes in the state, and grant them back to the original state as a hypothesis. The proof for such lemmas would be the two aforementioned consecutive lines of proof. This can lead to $(n-2)$ additional lemmas.

The *second backward path* generates lemmas where their proofs are the first $m$ lines of the original $n$ line proof. This leads to $(n-3)$ additional lemmas.

Therefore, in this case, one can theoretically extract $(3*n-7)$ lemmas. For a proof with 3 lines, this leads to extraction of 2 distinct lemmas. For a proof with 4 lines, this leads to extraction of 5 distinct lemmas, and so on.

This is a somehow mechanistic view of the process of breaking down the proofs. In practice, we have used human judgment. On average, for all the IMO problems in the paper, we have extracted about $n/3.5$ lemmas for $n$ lines of proof which is far less than the two bounds explained above.

The smallest ratio, in our dataset, is $n/5.6$ for IMO 1962 P2, and the largest ratio is $n/2.4$ for IMO 2022 P2.

## 5 Related work and a discussion on avoiding training/testing set contamination for better evaluation of model generalizations

One way to address the evaluation of generalization abilities of an AI model is to test them on challenging problems where the correct answer is not publicly available. For example, FunSearch (Romera-Paredes et al., 2024) works on combinatorial problems such as bin packing and comes up with a heuristic algorithm that is better than the best algorithm available in the literature. The last improvement on this problem was made decades ago, so it is clear that this algorithm was not copied from somewhere else.

AlphaGeometry (Trinh et al., 2024), on the other hand, evaluates its performance on all IMO problems in geometry from 2000 to 2022. To avoid training/testing set contamination, it trains its language model on a synthetic dataset generated automatically using an SMT solver. This method of using synthetic data addresses many of the evaluation concerns, but the starting point for creating the synthetic data is geometric figures, and then they exhaustively apply all the possible geometric lemmas to those figures. This approach works for geometric problems, especially when a capable SMT solver is available. Moreover, these geometric problems are decidable unlike the ones we study in this paper. For problems in number theory and algebra, the approach of exhaustively applying all the possible lemmas to a given state is not practical, as the number of applicable lemmas is much larger compared to geometric problems. Moreover, AlphaGeometry considers proving the problems in a system like Lean to be difficult and uses a more simple symbolic system developed by the mathematicians.

LeanDojo (Yang et al., 2023) observes that: "the common practice of splitting theorems randomly into training/testing has led to an overestimated performance in the previous papers. LLMs can prove seemingly difficult theorems simply by memorizing the proofs of similar theorems during training." To remedy this issue, it proposes a new data set in which the test set is designed to be challenging compared to the training set.

Outside the domain of formal mathematics, Skill-Mix (Yu et al., 2023) proposes an evaluation technique in which prompts are designed to include a combination of skills (such as metaphor, red herring, and common knowledge physics). Combining skills in this way aims to prompt the model to come up with a sensible answer that most likely does not already exist in the training set of the model. If the model were to produce

a sensible answer, it would need to draw from various parts of its training set. Skill-Mix's study provides positive evidence that GPT-4 can occasionally provide sensible answers to prompts combining a small number of those skills. But for most prompts, GPT-4 does not succeed.

Other studies use pretrained language models to prove theorems, i.e., LLMs that are trained on the contents of the Web. Some of these papers acknowledge that the answers to some of the testing problems might have leaked into the model training set, for example, First et al. (2023) reports: "there is the potential for proofs from the test set to have leaked into the LLM pretraining data. While the pretraining data for the Minerva LLM at the base of our models does not include the PISA dataset, it does contain code that may include some Isabelle/HOL proofs found in PISA. This should be kept in mind when interpreting the results."

Some studies do not try to make a clear separation between the training and tes sets; neither do they try to distinguish between the generalization and possible retrievals from the the training set or the prompt. For example, LEGO-prover (Wang et al., 2024) uses ChatGPT to convert a human written proof into a proof sketch for each of the problems in miniF2F, and then again uses ChatGPT to write the formal proof for each of the steps in the proof sketch. When prompting ChatGPT, it tags the possible relevant lemmas and their proofs in the prompt so that GPT can use them if needed. These additional lemmas are taken from a separate library. For each problem in miniF2F, LEGO-prover performs up to 100 proof attempts with ChatGPT costing around 300 dollars. At the end of the day, it is not clear how many of those success rates are randomly exhausting all possible combination of lemmas and how many of them might be direct retrieval from the contents of the prompts or contents of the ChatGPT's training set. Moreover, since ChatGPT is a model that evolves over time and it is not open-source, concerns about the reproducibility of the results arise.

## 6 Conclusion and Future Work

In this work, we provided the proofs for all the remaining IMO problems in the test set of miniF2F. We also presented a dataset of 1,329 lemmas as a stepping stone towards writing automated formal proofs for challenging IMO problems in number theory and algebra, derived from a subset of those IMO problems. We devised a systematic method to create a large data set of approachable lemmas formalized in Lean. These lemmas are the building blocks of proofs for IMO problems. They are considerably less challenging than IMO problems, yet state-of-the-art models struggle to prove most of them. Moreover, we performed an extensive human evaluation of o3-mini's responses to diagnose the issues this model encounters when it tries to write a Lean proof for these problems. Our experiments on these lemmas allowed us to diagnose and evaluate the obstacles towards proving the unsolved IMO problems in the miniF2F dataset, and beyond, including the more challenging ones from 2022 and 2023. Developing models that can prove the lemmas in our dataset would brings us closer to models that can participate and do well in IMO.

### Acknowledgments

Xuenan Cao's work is supported by a grant from the Research Grants Council of the Hong Kong Special Administrative Region, China, Project 14602223.

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

# A   Limitations

This work is a step towards writing formal Lean proofs for challenging problems in mathematics, specifically IMO problems. The IMO problems we chose cover some of subjects covered in International Math Olympiad, mostly the ones related to number theory and algebra. Topics such as combinatorics and geometry are not covered here. Moreover, our dataset does not cover all the topics under the category of number theory and algebra, though, we consider our work a major step in that pursuit.

Writing a proof for an IMO problem on number theory and algebra requires both planning and execution of the steps in the plan. One has to look both at the big picture and also at the building blocks that build on top of each other to create the proof. In this paper, our focus was largely on the building blocks and small lemmas, but to have an automated system that can prove such IMO problems one would need to give a considerable attention on planning a viable proof sketch and then putting the building blocks together.

Our dataset is in Lean. It may be converted to other languages such as Isabelle as well. We are familiar with Isabelle and we may perform this conversion if we see interest from the AI researchers who primarily work with Isabelle.

We do not foresee any negative societal impact specifically resulting from this work. Automation sometimes eliminates or reduces jobs. At the same time, it creates new jobs. Developing models and methods that can solve mathematical problems may have such effects on the labor market, but this is not necessarily a negative impact.

# B   Mathematical topics used in our dataset

Table B1 shows the topics used in each of the IMO problems we have studied in this paper. For an automated system to be able to prove the lemmas in our dataset, it will need to utilize these topics and concepts.

Table B1: Topics utilized in the IMO problems formalized and studied in this paper

| # | Problem | Types | Concepts used in the proof |
|---|---------|-------|----------------------------|
| 1 | 1959-P1 | Natural | divisibility |
| 2 | 1960-P2 | Real | algebra, inequalities |
| 3 | 1962-P2 | Real, Rational | quadratic discriminant and roots, quadratic factorization, algebra, inequalities |
| 4 | 1964-P2 | Real | algebra, inequalities |
| 5 | 1965-P2 | Real | linear system of equations, algebra, inequalities |
| 6 | 1983-P6 | Real | algebra, positive and negative inequalities, bounds |
| 7 | 1984-P6 | Natural | power inequalities, divisibility, algebra, prime factorization, greatest common divisor |
| 8 | 1985-P6 | Natural, Real, NNReal | functions, limits, infinite sequences, upper bounds and lower bounds, bijectivity, injectivity, monotonicity, continuity, sets, induction |
| 9 | 1992-P1 | Natural, Integer, Rational | factorization, primes, group theory, casting, exponential growth, absolute values |
| 10 | 1997-P5 | Natural, Real | prime factorization, divisibility, group theory, rings, logarithm, casting, induction |
| 11 | 2022-P2 | Real | algebra, inequalities, order, logic |
| 12 | 2022-P5 | Natural, Integer | congruence, primes, lifting the exponent lemma, factorial, finite sets, big operators, divisibility, functions |
| 13 | 2023-P4 | Natural, Real | finite sets, functions, big operators, casting, weighted geometric mean |

## C   Information about the formal proofs of IMO problems in our dataset

Table C2 lists the problem topic for each IMO problem and whether these proofs appeared in the MiniF2F dataset. We highlight that the majority of the released proofs were not available to the general research community.

Table C2: IMO problems formalized in this paper

| #     | Year | Problem | Topic          | in miniF2F | Lean proof publicly available | # of lines of Lean4 code |
|-------|------|---------|----------------|------------|-------------------------------|--------------------------|
| 1     | 1959 | P1      | number theory  | Yes        | Yes                           | 9                        |
| 2     | 1960 | P2      | algebra        | Yes        | Yes                           | 40                       |
| 3     | 1962 | P2      | algebra        | Yes        | No                            | 60                       |
| 4     | 1963 | P5      | algebra        | Yes        | No                            | 50                       |
| 5     | 1964 | P2      | algebra        | Yes        | Yes                           | 50                       |
| 6     | 1965 | P2      | algebra        | Yes        | No                            | 210                      |
| 7     | 1968 | P5      | algebra        | Yes        | No                            | 30                       |
| 8     | 1969 | P2      | algebra        | Yes        | No                            | 150                      |
| 9     | 1974 | P3      | number theory  | Yes        | No                            | 510                      |
| 10    | 1981 | P6      | algebra        | Yes        | No                            | 40                       |
| 11    | 1982 | P1      | algebra        | Yes        | No                            | 75                       |
| 12    | 1983 | P6      | algebra        | Yes        | No                            | 180                      |
| 13    | 1984 | P6      | number theory  | Yes        | No                            | 380                      |
| 14    | 1985 | P6      | number theory  | Yes        | No                            | 1,310                    |
| 15    | 1992 | P1      | number theory  | Yes        | No                            | 480                      |
| 16    | 1997 | P5      | number theory  | Yes        | No                            | 390                      |
| 17    | 2007 | P6      | algebra        | Yes        | No                            | 570                      |
| 18    | 2022 | P2      | algebra        | No         | No                            | 260                      |
| 19    | 2022 | P5      | number theory  | No         | No                            | 640                      |
| 20    | 2023 | P4      | number theory  | No         | No                            | 450                      |
| total |      |         |                |            |                               | 5,884                    |

# D    Information about the the extracted lemmas and their formal proofs

Table D3 shows statistics about the length of the proofs for all the lemmas that we have extracted from the IMO proofs, defining our lemma dataset. These lemmas are the building blocks of IMO proofs, and we have used them to evaluate the LLMs.

Table D3: Statistics about the length of the proofs of lemmas in our dataset

| #     | Problem | Mean | Max   | Min | Std  | Lemma count | Lines of Lean code for all lemmas |
|-------|---------|------|-------|-----|------|-------------|-----------------------------------|
| 1     | 1959-P1 | 2.3  | 3.0   | 1.0 | 1.2  | 4           | 30                                |
| 2     | 1960-P2 | 7.1  | 30.0  | 2.0 | 9.4  | 9           | 140                               |
| 3     | 1962-P2 | 6.6  | 16.0  | 1.0 | 4.7  | 14          | 220                               |
| 4     | 1964-P2 | 8.6  | 25.0  | 2.0 | 8.1  | 9           | 180                               |
| 5     | 1965-P2 | 17.2 | 158.0 | 2.0 | 29.6 | 73          | 2,950                             |
| 6     | 1983-P6 | 10.3 | 27.0  | 2.0 | 6.1  | 53          | 1,200                             |
| 7     | 1984-P6 | 13.4 | 137.0 | 1.0 | 22.2 | 64          | 1,620                             |
| 8     | 1985-P6 | 22.3 | 298.0 | 3.0 | 31.2 | 427         | 14,690                            |
| 9     | 1992-P1 | 11.0 | 70.0  | 1.0 | 12.5 | 91          | 2,135                             |
| 10    | 1997-P5 | 12.4 | 85.0  | 1.0 | 14.4 | 122         | 2,950                             |
| 11    | 2022-P2 | 12.2 | 66.0  | 1.0 | 14.5 | 61          | 1,630                             |
| 12    | 2022-P5 | 12.2 | 77.0  | 2.0 | 15.2 | 265         | 6,845                             |
| 13    | 2023-P4 | 22.2 | 115.0 | 1.0 | 29.5 | 137         | 5,580                             |
| Total |         |      |       |     |      | 1,329       | 40,170                            |

# E   Additional results on the langauge models

Table E4 presents the accuracy of all the LLMs that we have evaluated on our lemmas dataset. We include additional results on Llama 3.1 (Dubey et al., 2024) and Qwen2.5-Math (Yang et al., 2024) models. We note that these models were not explicitly trained on Lean 4 data but rather perform informal mathematical reasoning. Following this fact, they are unable to generate formal proofs for our lemmas dataset.

Table E4: Comparing performance of SOTA models on lemmas dataset

| Problem | # of lemmas | Deepseek Prover-v1.5-RL (@32) | Goedel-Prover (@32) | ReProver retrieval ✗ | ReProver retrieval ✓ | o3-mini (with 10 e.f.) | Qwen2.5-Math | LLama 3.1 |
|---|---|---|---|---|---|---|---|---|
| 1959-p1 | 4 | 3 | 2 | 2 | 2 | 2 | 1 | 0 |
| 1960-p2 | 9 | 7 | 6 | 3 | 4 | 1 | 0 | 0 |
| 1962-p2 | 14 | 13 | 12 | 7 | 8 | 6 | 0 | 0 |
| 1964-p2 | 9 | 5 | 5 | 5 | 5 | 3 | 0 | 0 |
| 1965-p2 | 73 | 48 | 47 | 47 | 46 | 12 | 0 | 0 |
| 1983-p6 | 53 | 25 | 32 | 28 | 29 | 18 | 0 | 0 |
| 1984-p6 | 64 | 32 | 33 | 25 | 24 | 13 | 0 | 0 |
| 1985-p6 | 427 | 116 | 116 | 89 | 89 | 84 | 0 | 0 |
| 1992-p1 | 91 | 48 | 54 | 35 | 34 | 25 | 0 | 0 |
| 1997-p5 | 122 | 51 | 49 | 48 | 51 | 30 | 0 | 0 |
| 2022-p2 | 61 | 34 | 30 | 24 | 25 | 25 | 0 | 0 |
| 2022-p5 | 265 | 89 | 76 | 80 | 77 | 60 | 0 | 0 |
| 2023-p4 | 137 | 52 | 41 | 43 | 45 | 37 | 0 | 0 |
| Total (abs) | 1,329 | 523 | 504 | 436 | 439 | 316 | 1 | 0 |
| Total (%) | 100% | 39.4% | 37.9% | 32.8% | 33.0% | 23.8% | 0.1% | 0% |

# F   Distribution of Proof Lengths in LLM-based provers

Figure F1 shows the distribution of the lengths of correct proofs generated by each LLM for the lemmas dataset. We highlight that all LLMs, except o3-mini, struggle to consistently generate lengthy proofs. DeepSeek-Prover-V1.5-RL and Goedel-Prover produce proofs reliably up to a length of 6, while longer proofs become increasingly rare. ReProver, both with and without retrieval, reliably generates proofs of length 3 or shorter. These figures illustrate that the current state-of-the-art models for theorem proving tend to focus on easier and shorter proofs, often overlooking longer proof chains that require planning and a structured understanding of mathematical statements. We note that distribution of human written proofs reflect a high number of problems that require proof chains that go beyond any of the evaluated LLMs. Training a model capable of solving lengthier and more complex proofs remains an ongoing challenge.

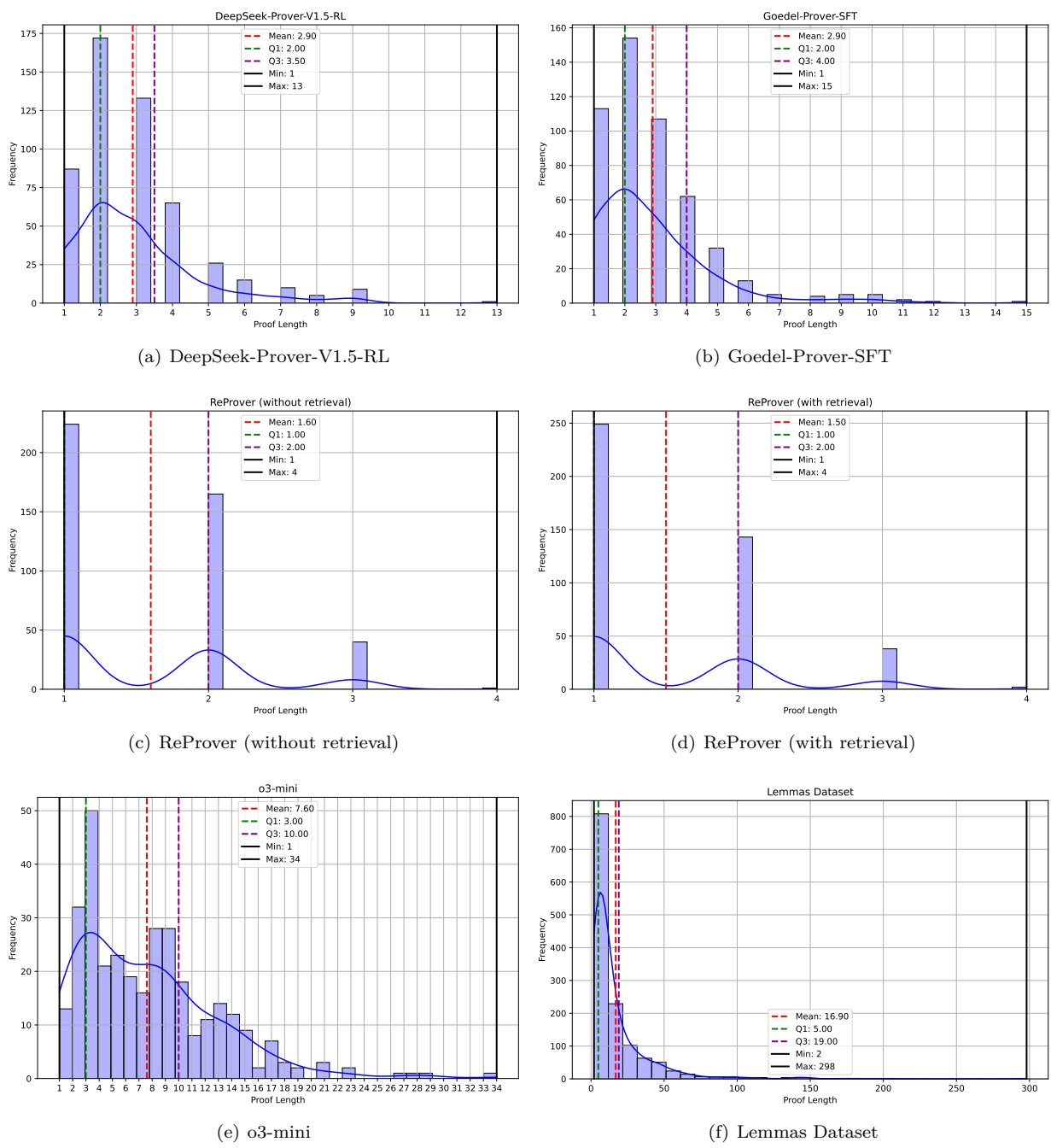

Figure F1: Distribution of proof lengths generated by various LLM-based provers on the lemmas dataset (a-e). Histogram (f) illustrates the distribution of length of human-written proofs.

# G    Redundancy in LLM generated proofs

In Section 3.3, we refer to proofs that contain unnecessary or redundant structures as "bloated." Figure G2 illustrates two cases of bloated proofs produced by LLMs. To better reflect the capabilities of modern theorem provers, we manually corrected such proofs when evaluating the length of generated proofs. This correction was done, merely, by commenting out the unnecessary lines, not by revising the proof sketch or by replacing existing with more compact lines.

In Figure G2(a), the proof can be concluded in a single line using *omega*; however, the LLM introduces an unnecessary structure before reaching this conclusion. In Figure G2(b), the LLM opts to prove hypotheses that are already provided as conditions in the formal statement of the problem. Instead of using them directly to close the goal, the LLM unnecessarily proves them again and then uses the newly derived hypotheses to conclude the proof. We count this as an unnecessary structure within the generated proof. The corrected proofs are presented to the right of the corresponding "bloated" proofs.

(a) Example #1

(b) Example #2

Figure G2: Diagrams that illustrate manual redundancy correction in LLM-generated proofs. The overall proof structure remains intact, with only unnecessary elements/structures removed.

## H  Datasheet

Following the framework in Gebru et al. (2021) and similar to Paster et al. (2023), Table H5 provides the data sheet for our dataset.

Table H5: Datasheet for our dataset

| Questions | Answers |
|---|---|
| **Motivation** | |
| For what purpose was the dataset created? | To provide a set of formal mathematical problems as a stepping stone for automated systems to prove challenging International Math Olympiad problems in number theory and algebra. |
| Who created the dataset and on behalf of which entity? | The authors of this work. |
| Who funded the creation of the dataset? | The companies where the authors work. |
| Any other comment? | None. |
| **Composition** | |
| What do the instances that comprise the dataset represent? | Lemmas formalized in Lean 4. |
| How many instances are there in total? | 1,329 lemmas. |
| Does the dataset contain all possible instances or is it a sample of instances from a larger set? | It is not a sample of a larger set. |
| What data does each instance consist of? | A formal lemma and its proof formalized in Lean. |
| Is there a label or target associated with each instance? | Not necessarily. |
| Is any information missing from individual instances? | No. |
| Are relationships between individual instances made explicit? | Not applicable. Each lemmas can be proved on its own, and it is not dependent or have an explicit relationship with other lemmas in the dataset. |
| Are there recommended data splits? | The dataset does not have such splits. However, it is possible for people to generate such splits. The authors do not advocate for such splits on this particular dataset. Instead, it is recommended to solely rely on the lemmas in the Mathlib library to prove the lemmas in this dataset. |
| Are there any errors, sources of noise, or redundancies in the dataset? | No. There are absolutely no errors in the dataset as all the lemmas and their proofs are automatically checked by the system of Lean theorem prover. Redundancies is hard to define in this context. There are no repeated lemmas in the dataset. |
| | Continued on next page |

| | |
|---|---|
| Is the dataset self-contained, or does it link to or otherwise rely on external resources? | It is self contained. It is written in Lean 4 programming language and it relies on the Mathlib library which is open source. |
| Does the dataset contain data that might be considered confidential? | No. |
| Does the dataset contain data that, if viewed directly, might be offensive, insulting, threatening, or might otherwise cause anxiety? | No. |
| **Collection Process** | |
| How was the data associated with each instance acquired? | 13 IMO problems were chosen as described in Table 1. The formal proof for each of these problems was written manually by the authors. These proofs were broken into their building blocks leading to a set of 1,329 lemmas written in Lean 4. The proofs for these lemmas were written by the authors. Some of the lemmas that can be proved by automated solvers in Lean were excluded from the dataset. The said automatic solvers are the following: *hint*, *linarith*, *exact?*, *simp*, *omega*, *ring*, *norm_cast* and *norm_num*. |
| What mechanisms or procedures were used to collect the data? | Dataset was created by the authors based on the mathematical proofs of 13 IMO problems listed in Table 1. |
| If the dataset is a sample from a larger set, what was the sampling strategy? | Not applicable. |
| Who was involved in the data collection process and how were they compensated? | Dataset was not collected. |
| Over what timeframe was the data collected? | Dataset was not collected. |
| Were any ethical review processes conducted? | No. |
| **Preprocessing/cleaning/labeling** | |
| Was any preprocessing/cleaning/labeling of the data done? | No. There was a post processing which excluded some of the lemmas that could be proved directly via automatic solvers in Lean. These lemmas were deemed easy to prove, and since existing automatic solvers can prove them, we do not consider them an obstacle in our larger goal of proving IMO problems. |
| Was the "raw" data saved in addition to the preprocessed/cleaned/labeled data? | Not applicable. |
| Is the software that was used to preprocess/clean/label the data available? | Lean prover was used to verify the correctness of the process which is an open source system widely used in the field. |

| | |
|---|---|
| Any other comments? | No. |
| **Uses** | |
| Has the dataset been used for any tasks already? | We have evaluated the ability of Goedel-Prover, DeepSeek-Prover-V1.5-RL, OpenAI's o3-mini, Reprover with and without retrieval, Qwen Math 2.5, and Llama 3.1 in proving the lemmas in our dataset. |
| Is there a repository that links to any or all papers or systems that use the dataset? | There will be a link on GitHub which will be public. |
| What (other) tasks could the dataset be used for? | The first goal would be to develop models and methods that can prove these lemmas in formal language and also in natural language. The second goal would be to develop models and methods that can use these lemmas to prove the IMO problems in this dataset as well as other IMO problems in number theory and algebra. |
| Is there anything about the composition of the dataset or the way it was collected and preprocessed/cleaned/labeled that might impact future uses? | We do not foresee such issues arising. |
| Are there tasks for which the dataset should not be used? | Not that we can think of. |
| Any other comments? | No. |
| **Distribution** | |
| Will the dataset be distributed to third parties outside of the entity on behalf of which the dataset was created? | Yes, we plan to release and host the dataset on GitHub. |
| How will the dataset will be distributed? | GitHub. |
| When will the dataset be distributed? | With the camera-ready version of the paper. |
| Will the dataset be distributed under a copyright or other intellectual property license, and/or under applicable terms of use? | Yes, we plan to release it under the MIT license. |
| Have any third parties imposed IP-based or other restrictions on the data associated with the instances? | No. |
| Do any export controls or other regulatory restrictions apply to the dataset or to individual instances? | No. |
| Any other comments? | No. |
| **Maintenance** | |
| Who will be supporting/hosting/maintaining the dataset? | The first author. |

| | |
|---|---|
| How can the owner/curator/manager of the dataset be contacted? | Email address and also via the available methods on GitHub. |
| Is there an erratum? | This is not necessary. All the lemmas and their proofs are checked and verified for correctness with the Lean system. |
| Will the dataset be updated? | Possibly. If a new version of Lean is released, e.g., Lean 5, we might need to update our dataset based on the new release of Lean and Mathlib. Occasionally, some lemmas in the mathlib library get modified/merged/renamed. Lean automatically marks such items in a proof. We will routinely go over the dataset and fix those issues whenever a new version of Lean is released. The current version of dataset is fully verified in Lean 4.17 which is the latest version. |
| If the dataset relates to people, are there applicable limits on the retention of the data associated with the instances? | Not applicable. |
| Will older versions of the dataset continue to be supported/hosted/maintained? | Yes. |
| If others want to extend/augment/build on/contribute to the dataset, is there a mechanism for them to do so? | Yes. There are two avenues. First, the dataset written in Lean can be rewritten in other languages such as Isabelle. Second, more problems (IMO or otherwise) can be formalized in Lean and/or other languages, and their proofs can be turned into a set of additional lemmas to expand the current dataset. |
| Any other comments? | No. |

## I    License and authors' responsibility statement

We plan to release our dataset under the MIT license on GitHub and also via Croissant. Authors bear all responsibility in case of violation of rights.

## J    Other motivations

Earlier, we outlined the motivation for developing AI models that can perform well in IMO. We have noted some of the other immediate and practical motivations for building such systems. Here, we elaborate on what has been delineated above.

We have mentioned Amazon's use of Lean system for formal verification of software. Besides Amazon, Lean is being used by a rather large community of mathematicians to write formal proofs for a variety of mathematical problems. For example, Terry Tao and a team of other mathematicians are currently formalizing the prime number theorem (i.e., the asymptotic distribution of the prime numbers among the positive integers) in Lean. For an ordinary theorem, it is reported that writing a formal proof in Lean may take up time more than 10 times longer than writing the same proof with pen and paper. For the more complicated topics such as the prime number theorem, it may take months to build the necessary background in Lean language to prove the ultimate theorem. However, the correctness of a proof written in Lean can be verified instantly, but the proof with pen and paper may have shortcomings or even mistakes that are hard to detect. For complicated problems, finding a human reviewer to verify a proof written with pen and paper may not be easy.

As the Mathlib library grows, and as more lemmas become available in the library, lemmas that can be utilized in writing new proofs, the amount of time for writing a proof for simple problems is expected to reduce. In writing these proofs, mathematicians use their knowledge of Lean and Mathlib to write the proofs. Machines may also be helpful in reducing the time for the formalization of proofs (Buzzard, 2024). In Lean, there are automatic methods that can suggest tactics or even prove some theorems by applying existing lemmas. Researchers also developed LLMs that can suggest tactics aiming to help mathematicians at writing the formal proofs, e.g., the Copilot by Song et al. (2024). However, such LLMs appear to be in their infancy, e.g., Copilot only proves easy lemmas that are designed for new learners of Lean.

On the other hand, for mathematicians, there are always harder theorems and conjectures that can be taken up to be formalized. In other words, there is no limit on how large a mathematical library can grow. Most recently, Kevin Buzzard started a project to formalize the proof for Fermat's last theorem in Lean, and this project is expected to run for a few years (Buzzard & Taylor, 2024).

The process of peer review in these fields of mathematics is largely concerned with verification of correctness of proofs. So, when a new paper is written and its proofs are formalized in Lean, the review process becomes much easier and faster. The large memory and computational power of computers can also be beneficial in proving novel theorems that require vast exploration of possibilities and hard for a human to prove and verify.

In summary, having automated systems that can follow a certain logic, utilize a library of existing lemmas, and prove novel mathematical theorems would have a significant impact on mathematical research and possibly other fields such as design of algorithms, formal verification, etc.

