# OpenReview forum: "A Lean Dataset for International Math Olympiad: Small Steps towards Writing Math Proofs for Hard Problems"
_TMLR — Accepted by TMLR_

### Review · Reviewer_oHDK · 2024-12-11

**Summary Of Contributions:**

This paper presents a LEAN dataset containing formalized solutions to IMO problems, including those from the miniF2F dataset and recent IMO competitions (2022, 2023). These problems are decomposed into over 900 smaller lemmas, totaling more than 25,000 lines of LEAN code. The work then evaluates the ability of o1-mini to write formal proofs for these lemmas using zero-shot prompting, chain-of-thought (CoT) reasoning, and lemma retrieval. This work aims to create a stepping stone for AI models to write proofs for challenging IMO and miniF2F problems and beyond.

**Audience:**

Yes

**Claims And Evidence:**

Yes

**Requested Changes:**

Requested Changes
1. Expand the dataset to include more coverage. For instance, include geometry and combinatorics to ensure comprehensive training and evaluations. Including more diverse mathematics topics would be greatly beneficial.

2. Develop the feasibility/infeasibility of semi-automated tools for lemma decomposition that require significant computational resources to reduce manual effort and improve scalability.

3. Explore approaches for harder-to-formalize problems to facilitate better generalization in models, enabling them to solve a wider range of difficulty levels.

**Strengths And Weaknesses:**

*Strengths*

1. The novelty of creating a comprehensive dataset in LEAN offers a **significant** resource and advancement in automated theorem-proving research.
2. By breaking down complex problems into lemmas, this paper provides greater transparency in decomposing lemmas, making them suitable for LLM evaluation and training.
3. The dataset's adaptability to other formal proof systems, such as Isabelle, makes it versatile for future research.

*Weaknesses*
1. The current methods may not scale efficiently to handle larger, more diverse, or harder datasets, which require significant computational resources due to reliance on manual lemma decomposition.
2. The evaluation focuses heavily on o1-mini without considering other potential models or frameworks that could address the challenges more effectively, limiting the scope of comparative analysis.
3. The paper does not address or acknowledge problems that may be unsolvable by LEAN. Some Olympiad-based questions may not be solvable using symbolic theorem-proving methods yet.

---

> ### Author Response · Authors · 2025-01-05
>
> Thank you for your review of our paper and your suggestions. Please see our responses using the same numbering in your review.
>
> -------
>
> #### **Requested changes:**
>
> 1. We have limited the scope of this paper to the scope of the miniF2F dataset which is the benchmark of the community. We are not planning to consider other mathematical topics in this paper. Other topics such as combinatorics and geometry require a more focused attention. If we add those other topics in the same paper, the length of the paper will be much longer than the standard of the community, and it will also require several months of extra work. To justify our decision, we give some examples from the literature:
>
>     a. The miniF2F dataset and all the papers that only evaluate on miniF2F have the same subject scope as our current paper.
>
>     b. The Alpha Proof model from Google DeepMind only proves problems on number theory and algebra, same as our paper and same as the miniF2F.
>
> 2. Our lemma decomposition in this paper is not automated, and as we have explained, it is human made. Automating this process is an interesting research direction that falls outside the scope of the current paper. This decision is again based on the contents of the current paper and our view that automating the decomposition process and evaluating its implications deserves a dedicated paper.
>
> 3. We are not entirely sure what the reviewer means by “harder-to-formalize problems”. If the reviewer has a specific problem (from miniF2F or from IMO) in mind, that they consider harder than what we have formalized, we would be happy to hear it.
>
> --------
>
> #### **Weaknesses:**
>
> 1. We are not sure what the reviewer means by “larger, more diverse, or harder datasets”. We are aware of a new dataset in the community, PutnamBench, which was recently published at NeurIPS 2024. Is that the dataset that the reviewer has in mind? That dataset only has the problem statements without their formal proofs, and the reported ability of LLMs to prove the problems in that dataset is very low. So, our approach for that dataset would also be to try to formalize them using a combination of human effort and available automated systems. And of course, once the proofs are available, one can try to decompose the proofs into their building blocks, and evaluate the ability of models to prove those building blocks, too. We do not argue that such evaluation should be universal in the community, but comprehensive evaluations on the benchmarks of the community provides useful insights, in our opinion.
>
> 2. Previously, we evaluated the ability of GPT-4 on our lemmas and we have the complete results. The accuracy of GPT-4 was close to zero for all the IMO problems after the 1960s. The ability of o1-mini is considerably better than GPT-4. We chose the o1-mini as the most advanced model that does not take too long to answer, making it practical to perform a large scale set of experiments on more than 900 lemmas. We can include the results of GPT-4 in an appendix.
>
> 3. We are not sure what olympiad problems cannot be proved in Lean. Mathlib, and not necessarily the Lean language, has some specific definitions, for example, division by zero yields zero; or for natural numbers, subtractions that lead to a negative number will yield zero, because a natural number cannot be negative. Incorporating these in formalizing a problem might sometimes require adding careful constraints in the problem statements, but in our view, it does not make the formalization of such problems impossible. For some other topics, formalizing a proof might require developing a relatively large library of basic definitions and lemmas. We have personally encountered such problems when formalizing problems on integration, combinatorics, and asymptotics. Again, this does not make the formalization of a problem *unsolvable* or impossible. We will add a discussion on this topic in the appendix. We appreciate your questions and comments.

---

> > ### Comment · Reviewer_oHDK · 2025-01-07
> > **Response to Rebuttal**
> >
> > 1. Thank you for your clarification. Regarding the Minif2f dataset, it primarily consists of questions within algebra and number theory. I’m particularly curious about how it performs with other varieties of mathematical problems, such as combinatorics and graph theory. Does it generalise well to questions outside algebra and number theory? For instance, how does it perform on the MATH dataset (Level 4/5), which includes combinatorics, number theory, algebra, and other areas? While Minif2f particularly targets AIME, IMO, and MATH datasets for number theory and algebra, it would be insightful to explore its performance in broader mathematical contexts.
> >
> > 2. Thank you for your clarification. For further clarification, does GPT-4's performance include GPT-4o, or is GPT-4o excluded? It would be interesting to know if GPT-4o also scored zero on IMO problems.
> >
> > 3. Please refer to section 1 for questions outside the scope of Minif2f. It would be a huge contribution if the dataset could be formalised to cover a wider range of mathematical problems. Clarifying this in the appendix would be helpful.
> >
> > Overall, the paper is very interesting and has great potential to contribute to advancements in theorem provers!

---

> > > ### Author Response · Authors · 2025-01-27
> > >
> > > Thank you for your feedback, your suggestions, and for acknowledging the contributions of our work.
> > >
> > > To address your suggestion, we are in the process of adding the evaluation of ReProver (with and without retrieval) on our dataset as well as the results on GPT-4. Our experiments are on GPT-4, not GPT-4o. We believe with the additional experiments, our results section will adequately represent the abilities of models currently available.

---

### Review · Reviewer_6z2g · 2024-12-15

**Summary Of Contributions:**

This paper addresses a gap in the automated theorem proving community by providing formal Lean proofs for previously unformalized International Mathematical Olympiad (IMO) problems in the miniF2F dataset. The key contributions are:

The authors formalize 12 IMO problems that lacked formal proofs in Lean, including 9 from the miniF2F dataset and 3 recent problems from IMO 2022-2023. This expands the available formal proofs by contributing 5,150 new lines of Lean code.
They decompose these proofs into approximately 900 smaller lemmas totaling 25,500 lines of Lean code, creating a dataset that can potentially serve as intermediate steps for evaluating theorem-proving models.
They evaluate the o1-mini model's ability to generate formal proofs for these lemmas through zero-shot prompting, chain-of-thought reasoning, and providing feedback. Their systematic analysis reveals significant limitations in current models' capabilities for formal theorem proving.

**Audience:**

Yes

**Claims And Evidence:**

Yes

**Requested Changes:**

See weaknesses section

**Strengths And Weaknesses:**

Strengths:

Fills a significant gap by providing formal Lean proofs for the (most) widely used miniF2F dataset, enabling researchers to better evaluate and develop theorem-proving models.


Weaknesses:

1  Misaligned Research Contribution
The paper's approach of creating static proof decompositions is fundamentally mismatched with how AI models actually generate proofs. Since models produce dynamic, varied solutions, a static dataset of lemmas and predetermined decompositions offers limited practical value for improving automated theorem proving systems.


2  Imbalanced Content Distribution
The paper spends excessive space on peripheral details (like extensive proof descriptions and theorem statements) while lacking substantial content about core contributions.


3  Presentation issues: Incorrect terminology (using "native Lean" instead of "tactics") and inconsistent formatting in section titles  ("Hinting and Chain of Thoughts")

---

> ### Author Response · Authors · 2025-01-05
>
> Thank you for your kind review of our paper. We appreciate your thoughtful summary. Below we respond to each of the items under Weaknesses in your review.
>
> * The critique of “Misaligned Research Contribution”, we believe, depends on how one might want to use our dataset. Our dataset provides a stepping stone for writing complex formal proofs for IMO problems. This stepping stone is by way of evaluation, and not by way of direct training. Please see our general response above regarding this. We will be adding more discussions and will adjust the framing of the introduction to make this clear.
>
> * What the reviewer considers peripheral and substantive can be subjective, but it is helpful to know how a member of the community views the discussions in our paper. We hope that the discussions that we will add to the paper, especially the ones in response to the reviewer LYC8‘s suggestions will address your concern.
>
> * We will adjust some of the wordings in section 2, but we respectfully disagree with you on the correctness of our terminology. If we replace the term “tactic” in our paper, most of those sentences will be wrong. Tactic is a very broad term in Lean language that refers to most items that might appear in a proof. Terms such as exact, apply, and refine, are all tactics in the Lean language. Almost everything that appears in our proofs are tactics. In section 2.2, we have clearly listed the set of automated native tools in Lean that we have considered for excluding very easy lemmas. These tools are: hint, linarith, exact?, simp, omega, ring, norm_cast and norm_num. These methods automatically apply a set of procedures trying to close the goal, or to make an incremental step. What is common among these are that they are automated. If any of these commands automatically proves a lemma, we exclude that lemma from our dataset.
>
> * We adjusted the formatting issues. We appreciate your helpful comment.

---

> ### Comment · Reviewer_6z2g · 2025-01-06
> **Thanks for the reply**
>
> Thank you for your reply. While I understand most of your points, I'm not yet convinced to change my perspective on this work's motivation and contribution (which I still consider weak).
>
> I'm particularly unclear about your distinction regarding "native Lean." I have two main concerns:
>
> 1  I haven't encountered papers using this term to describe tactics like linarith, ring, etc. If you've seen such usage, could you please provide references?
>
> 2  After reviewing section 2.2 again, I'm uncertain about the rationale for excluding these tactics. While they are more sophisticated than basic tactics like 'exact' and thus reduce the burden on LLMs, determining when and how to apply them involves important reasoning skills that are central to automated theorem proving. Furthermore, I'm not aware of other research that deliberately limits itself by avoiding such proofs.

---

> > ### Author Response · Authors · 2025-01-27
> >
> > Thanks for your follow up questions.
> >
> > 1. We plan to use the term “automatic solvers in Lean”. This is the rewording that we had in mind mentioned in our previous response. We are not aware of any reference for this term, or any alternative term – we had to come up with this term. The term might be unfamiliar, but in the paper, we clearly define it by listing the tactics/solvers. This term only appears on 3 occasions in the body of the paper, and we believe those occurrences are close enough to where we define the term. In the data sheet, we will repeat the definition.
> >
> > 2. We do not limit our proofs to avoid those tactics. Section 2.2 does not make such a statement. Our approach is that we do not include any lemmas that can be proved automatically inside the Lean system using those solvers/tactics. This eliminates lemmas that might be considered too easy to prove, and defines a floor for the easiness of the lemmas. For example, consider a lemma that can be proved ONLY by writing a *linarith* as its proof in the Lean system – such a lemma is not included in our dataset. When breaking down a proof into its building blocks, it makes sense to define a floor on the easiness and size of the building blocks. If no floor is defined on the easiness, one might end up with an overwhelming number of lemmas, some of them as trivial as proving 1 < 2. We will explain this in more detail in the paper.

---

### Review · Reviewer_LYC8 · 2024-12-28

**Summary Of Contributions:**

The paper makes three contributions. The first is the formalization of 16 IMO problems that have not previously been formalized in Lean (13 from miniF2F's test set and 3 from recent IMOs). The second is the mechanistic/automated extraction of a set of 900 non-trivial lemmas from a subset of these 16 problems. The third is a study of OpenAI's o1-mini's ability to suggest and correctly formalize proofs for these lemmas.

**Audience:**

Yes

**Broader Impact Concerns:**

None.

**Claims And Evidence:**

No

**Requested Changes:**

- If the main contribution of the paper is to be the formalization of the 16 IMO problems, which I recognize is an extremely laborious process which the authors surely put a lot of time and effort into, the paper needs to discuss the concrete implications of this work in more detail. For example, please discuss what steps you would take to mitigate the ensuing risk of contaminating the miniF2F test set.
- If the main contribution of the paper is to be the set of 900 lemmas you extract from your handwritten proofs, the paper needs to be re-framed so as to clarify why this contribution is useful to the community. For example, you could do a thorough evaluation of how current/existing methods perform on this task, or you could look at whether using this dataset in some way (e.g. by fine-tuning on it) improves performance on other tasks, etc.
- For table 2, maybe there is a more useful column than the % of problems where the NL and Lean proof "match"? I think there's a lot confounding variables here that makes this statistic hard to interpret; for example, in line 2, where you report that 78% of the lemmas have correct informal/NL proofs but only 25% are actually formalized correctly, the agreement between the two being 89% (which far exceeds 25/78) seems informative; but in line 12, all of the disagreement seems to already be captured in whether the proof was correct or not, since \(15/28 \approx 0.53\) (assuming some rounding). Maybe there is an alternative way to present the statistic that clarifies what you are trying to communicate to the reader.
- Please rectify the typos mentioned above.

**Strengths And Weaknesses:**

Strengths:
- The paper is clearly written.
- Formalizing IMO problems is notoriously laborious and challenging; especially more recent problems, as evidenced by the several-hundreds-of-lines long proofs contributed by the authors. This type of work is ultimately crucial for advancing the field, slowly chipping away at the data scarcity plaguing AI4Maths.
- The lemma dataset constructed *might* be useful for future work in this area.
- I appreciate the honest discussion of the strengths and weaknesses of future work; for example, the notes on page 4 about mistakes made in the evaluation of some other recent contributions. Errare humanum est; this sort of honest discussion is very important to maintain a healthy research community.

Weaknesses:
- I'm not entirely convinced about the motivations of this paper. The authors spend a great deal of time alluding to data contamination issues in current evaluation sets; how does publicly listing formalized answers for 13 of miniF2F's *test* set problems help with that? If anything it will only make the matter worse, since from now any model trained on internet-scale data will contain contaminated data for miniF2F.
- The experiments are severely lacking in detail. For example, how many samples/attempts did you average over? What temperature was used for decoding? Looking at the appendix it appears that the experiments were conducted by interacting with o1-mini through the web UI, which I do not think makes for an adequate setting for such experiments.
- I believe the strongest contribution of this paper is the set of 900 lemmas you extract from your handwritten proofs. Unfortunately, the subsequent discussion thereof is very limited in scope; usually when contributing a new dataset it is customary to benchmark a range of existing methodologies on it, so that the reader can get a sense of what this new dataset might be used for. As it stands, I really have no clue; for example, I don't know whether the lemmas would transfer to new problems (so that I should consider finetuning my models on them, for example), whether they expose gaps in the scope of current systems that we as a community should study further, etc.
- This is a much more minor weakness, but there are many typos and mistakes made in the writing, e.g.:
  - Section 1.1 repeats itself over and over again; it could be made significantly more succinct
  - Bottom of page 2: "which is the first problem of the first IMO problem", should probably be "of the first IMO *competition*"
  - Page 3: "lemma decomposition on them": drop the "them"
  - Page  5: There's a broken section reference, listed simply as "??"
  - Page 7: "number lemmas" should be "number of lemmas"
  - Section 5: Wrong citation command used throughout (author names should be wrapped in parentheses here)

---

> ### Author Response · Authors · 2025-01-05
>
> Thank you very much for your thoughtful review of our paper. We really appreciate your feedback and suggestions.
>
> In our response, we use the same order as the bullet points in your review.
>
> -----------
>
> #### **Requested changes:**
>
> * 1 and 2 . Please see our general response. We will reframe the introduction section and will add the discussions that you have requested, specifically, more details about the implications of our work.
>
> Moreover, regarding item 1 and the concern about the proofs being used in the training set of LLMs, we think that, overall, releasing the formal proofs will help the community move forward. Once the proofs become available to the community, many models might be trained on them, and many models may become able to reproduce the correct proofs for the IMO problems in the test set of miniF2F. We think this milestone is inevitable, and once the community reaches that point, the gained accuracy will be for everyone, and not just for closed source models such as Open AI’s, or models with unclear training sets such as DeepSeek and Llama. Beyond that point, we think that the community might turn its attention to the training set of models, and not just the reported testing accuracy of models on miniF2F. A contribution in the field would then require a paper to show that their model is proving those IMO problems while their training set is not contaminated, because most models trained on the internet data will achieve very high accuracy on those problems and that becomes uninteresting to report. If you have any additional thoughts or suggestions on this, we would love to hear.
>
> * Regarding item 3, we are considering adding another column to the table to make the interpretation more insightful. We will certainly add such information in the paper.
>
> * Regarding item 4, we have now corrected all the items in your list, and will do an additional proofreading for the final version of the paper. Thanks again for your helpful comments.
>
>
> ---------
>
> #### **Weaknesses:**
>
> * We hope adjusting the framing of the introduction and the discussion in our general response (above) will address this concern.
>
> * We have conducted our experiments via the web interface. When we conducted our experiments back in October and November, o1-mini was available via API only to Tier 5 users, the most expensive Tier that we are not a member of. As far as we know, in the literature on ATP, a certain number of attempts are allowed for a given model to prove a problem, and the result is either failure or success within that number of attempts. This is the standard that has been followed in many papers in the literature with no averaging.
>
> * We think the discussion in our general response, that will be added to the paper, addresses this concern. We do not suggest that models should be fine-tuned on the formal proofs that we provide nor on the lemmas in our dataset -- although some might consider performing such fine-tuning. We expect the test set of miniF2F to still be used as a test set. Therefore, we suggest our lemmas be used for evaluation purposes, specifically, to understand and diagnose the failures of a model. Consider the case when someone reports to you that a LLM has failed to write a proof for IMO 2022 P5. This is a binary result and you would not even know how many lines of Lean proof are needed for this problem. On the other hand, consider that it is reported to you that the proof for IMO 2022 P5 has 640 lines, decomposed into 266 lemmas, and the accuracy of LLM on those lemmas is 17%. You may be able to understand and diagnose the failures of that LLM better than the first case where you were provided with a binary result of failure. Similarly, in formal verification, a SAT solver might fail in a given instance, but the failure itself does not provide adequate insights about the satisfiability of the problem.

---

> > ### Comment · Reviewer_LYC8 · 2025-01-07
> >
> > Thank you for your reply.
> >
> > I believe your take on the data leakage concerns, and the use case you have imagined for your lemma dataset, to be reasonable. I'm looking forward to reading an updated version of the manuscript in which you have clarified these points, so that it is clear to the reader what use your contribution may be to them.
> >
> > However, I must say that with your diagnostic use case in mind, I really find the experiments/analysis of the paper to be insufficient. I understand your reasons for interacting with o1 through the web interface, but there are many other models (some of which can even be run locally on consumer-grade hardware, and many others - like Gemini - that would not have been expensive to run) that you could have analyzed, too. These models are out there in the open - why put the onus on the reader to run your benchmark themselves even if they are simply using one of the currently available off-the-shelf models? As it stands, I do not believe you have given sufficient evidence for any reader of the paper to believe that your dataset is, indeed, a useful diagnostic tool.
> >
> > Finally, regarding averaging over multiple samples, there is a clear difference here to the familiar ATP setting, in which the prover is usually deterministic. Now, the "prover" (LLM) you are using is inherently a stochastic program. Thus the usual caveats and requirements of good statistical practice apply, and your results hence do not mean much if they are based on single samples.

---

> > > ### Author Response · Authors · 2025-01-27
> > >
> > > Thank you for your additional feedback.
> > >
> > >
> > > We are in the process of adding the evaluation of ReProver (with and without retrieval) on our dataset. We will also include the performance of GPT-4 on our lemmas which we had conducted previously. The accuracy of GPT-4 on our lemmas was very low for the lemmas extracted from IMO problems after the 1960s. The accuracy of o1-mini on our lemmas is way higher than any model we have encountered. That is why we decided to do the harder work of evaluating the accuracy of o1-mini.
> > >
> > > Clearly, more models can be evaluated on our lemmas. We believe after adding the results of GPT-4 and ReProver (with and without retrieval), the results in our paper will be representative of the variety of models currently available. The deep analysis of the results that we have provided: labeling various types of mistakes, evaluating the proofs in natural language, etc, are unprecedented in the literature, and we hope the reviewer does not overlook them.
> > >
> > > About the number of models used for evaluation, we give three examples from the literature:
> > >
> > > - LEGO-Prover from ICLR2024 evaluates on ChatGPT
> > >
> > > - [1] from ICML2024 evaluates on GPT-4
> > >
> > > - PutnamBench from NeurIPS2024 evaluates on GPT-4, COPRA, and ReProver
> > >
> > > -----------
> > >
> > >
> > > Regarding your suggestion about averaging the accuracies and the stochasticity of the output of LLMs, we understand your point, and we do not disagree with it. In our paper, we are not introducing a new method of evaluation regarding the use of LLMs in ATP. For the evaluation of ReProver, we run their own evaluation code with no modification. For evaluating o1-mini and GPT-4, we have also followed the same method of evaluation. ReProver (i.e., the LeanDojo project) also uses a LLM.
> > >
> > > We think that there are some justifications for the common practice in the community for not averaging when reporting the ATP accuracies with LLMs. One reason is that multiple rounds of prompting are performed. This evaluation practice goes back to [2] as far as we know, adopted later by [3] and many other papers, and its goal is to see how many theorems can a LLM prove after a certain number of attempts.
> > >
> > > Nevertheless, we agree with you that issues about the stochasticity in the responses of LLMs for ATP is an interesting topic that can be the subject of a specific study. For the current paper, though, we are following common practices in the literature for evaluation of LLMs for ATP.
> > >
> > > -----------
> > >
> > >
> > > We are working on the revision of the paper and plan to submit it soon. Thank you again for your suggestions and feedback.
> > >
> > > ---------
> > >
> > > [1] Murphy, L., Yang, K., Sun, J., Li, Z., Anandkumar, A. and Si, X., Autoformalizing Euclidean Geometry. ICML 2024.
> > >
> > > [2] Polu, S. and Sutskever, I., 2020. Generative language modeling for automated theorem proving. arXiv preprint arXiv:2009.03393.
> > >
> > > [3] Polu, S., Han, J.M., Zheng, K., Baksys, M., Babuschkin, I. and Sutskever, I., Formal Mathematics Statement Curriculum Learning. ICLR 2023.

---

### Author Response · Authors · 2025-01-05
**General response**

Dear Reviewers,

We think that it would be helpful to add a general clarification which relates to all the reviews.

Our approach in this paper is to provide a stepping stone for proving IMO problems in the miniF2F dataset and beyond, as the paper’s title suggests. We agree with the reviews that we need to clarify what this approach entails. The stepping stones that we provide are aimed at evaluation purposes with some implications for training.

An illustration might be helpful here. For example, if a teacher encounters a student who cannot solve a hard problem, the teacher might ask the student to solve easier problems first, such as the building blocks of that hard problem. How much the student can solve the easier problems will help the teacher evaluate the student’ strength and weaknesses. Likewise, the building blocks we provide in our dataset can be considered stepping stones for evaluation purposes. For further illustration, we can also break down the evaluation into scenarios such as the following:

* The student cannot provide a correct answer for any of the easy problems. The teacher might start asking even simpler questions.

* The student can provide a correct answer for some of the easy problems. The teacher might be able to identify the underlying weaknesses of the student. Perhaps the student has not been exposed to certain subjects. Or, maybe the student has not learned certain reasoning methods.

* The student can solve all of the easy problems. Then the teacher might try to teach the student how to put the answers for the easy problems together to compose the answer for the more complex problems.


In the case of our paper, we are addressing a similar situation that the community is facing. We have a benchmark called miniF2F, but models cannot prove most of the IMO problems in its test set. The community also does not have the formal proofs for most of these IMO problems. When an AI model fails to prove an IMO problem in the miniF2F dataset, we can be the teacher in the example above. We can ask the AI model to prove some related but easier problems, to evaluate its abilities and to diagnose its failures. Based on how the AI model performs on those easy problems, we can evaluate the weaknesses of the model better and make a better decision on how to improve the model.

The lemmas that we provide in our dataset are analogous to the easy problems that the teacher uses in the example above. So, our lemmas are aimed at better evaluation of models and to provide a way to diagnose their failures. Clearly, the results on the o1-mini model indicates that the model is still far behind the ability to achieve 100% accuracy on all the lemmas in our dataset. If a model achieves that accuracy on our lemmas, it may still fail to write a complete proof for the IMO problems.

Of course, the results of such evaluation will have implications for training as well. But, for an AI model to gain the ability to prove the lemmas in our dataset, it does not necessarily have to be trained on the exact same lemmas – that would not be considered generalization. Many components may be required to create such generalization ability in an AI model including the model architecture and model procedures, especially with regard to reasoning, and also, curation of appropriate training sets.

Seeing the performance of o1-mini on our lemmas is arguably more insightful than merely reporting that o1-mini proves 4 out of the 20 IMO problems in the miniF2F dataset.

One may argue that AI models, and LLMs in particular, do not prove a math problem the same way that a human does. If we agree with that view, our lemmas can still be interesting, especially to understand how an AI model works. As we have reported, o1-mini struggles to prove many of the lemmas in our dataset, and that means there is still room for improving this model. Hypothetically, if a model can prove an IMO problem, but it cannot prove its building blocks, i.e., our lemmas, this would be an interesting way to evaluate how this happens and whether that is an indication of memorization as opposed to reasoning and generalization. We have not encountered such cases though.

---

### Decision · Action_Editor_Tb2S · 2025-01-28

**Recommendation:** Accept with minor revision

**Comment:**

Dear authors,

I have gone over the reviews after a lengthy discussion between the reviewers and everybody involved, including yourselves.

I have recommended "minor revisions acceptance", noticing that in their final decision and throughout the discussion, the reviewers' most immediate concern was experimentation with LLMs and your dataset rather than presenting a significant conceptual flaw.

Specifically, an issue echoed throughout the reviews is that given that this is a new benchmark for mathematical reasoning, one would expect a much more thorough empirical investigation on how complex the dataset is and what stronger models can do with it. There was also a smaller concern about the focused domain in which you provided the dataset, although this is less of an issue to me, and I do not expect to be corrected should you decide to resubmit.

The amount of work that was required to formalize the proofs in your dataset, I am sure, is valuable, and I hope you will be able to improve the empirical evaluation part and bring this dataset to light in the LLM community.

In the revised version, I would expect an evaluation of your benchmark (as requested by the reviewers) on gpt-o1, Qwen2.5-Math and a llama3 model (for example, llama-3.1 or possibly even 3.2 or 3.3 if you have the resources). There was also a chat about DeepSeek-R1, given its recent advancement. We agreed it was desirable but not necessary to present results for it, given the timeline you submitted the paper.

It is also important to emphasize that your dataset should be considered as a benchmark rather than a way to train models (but you clarify that).

The AE

**Audience:**

The main audience for this paper would be people who work on mathematical reasoning with LLMs, and possibly LLM reasoning in general.

**Claims And Evidence:**

The paper, as it stands, supports its claim about the introduction of the presented dataset as a useful dataset for benchmarking mathematical reasoning. More work, as detailed below, is required for final acceptance.